# Heart Rate Variability in Elite International ITF Junior Davis Cup Tennis Players

**DOI:** 10.3390/biology12010017

**Published:** 2022-12-22

**Authors:** Santos Villafaina, Miguel Crespo, Rafael Martínez-Gallego, Juan Pedro Fuentes-García

**Affiliations:** 1Facultad de Ciencias del Deporte, Universidad de Extremadura, Avda: Universidad S/N, 10003 Cáceres, Spain; 2Departamento de Desporto e Saúde, Escola de Saúde e Desenvolvimento Humano, Universidade de Évora, 7005-849 Évora, Portugal; 3Development Department, International Tennis Federation, London SW15 5XZ, UK; 4Department of Sport and Physical Education, University of Valencia, 46010 Valencia, Spain

**Keywords:** performance, anxiety, sport, autonomic modulation

## Abstract

**Simple Summary:**

The precompetitive anxiety should be controlled in athletes to optimize their performance. In this regard, heart rate variability (HRV) is a non-invasive tool to assess the autonomic nervous system balance. Therefore, this study aims to investigate the HRV response of elite international junior tennis players during an international tournament. Eleven players participate in this study, with a mean age of 15.36 (0.67) years old. The HRV of tennis players are assessed 24 h before the event, and immediately before the first and the second match of the ITF Junior Davis Cup event. Each of the registers lasted 5 min and the time domain, frequency domain, and non-linear variables were extracted. Results show that elite tennis players did not decrease the HRV between the baseline and the first match. However, the HRV response significantly differed between the baseline and the second match. Nevertheless, anxiety was higher before the first match than before the second match. Coaches and physical trainers could use HRV to control and manage the precompetitive anxiety of junior tennis players. The experience in international tournaments and the familiarization with the environment in the second match could explain these results.

**Abstract:**

The aim of the present study was to investigate the evolution of the HRV during an international team tennis tournament in elite junior tennis players. A total of 11 tennis players, with a mean age of 15.36 (0.67) from six nationalities, participated in this cross-sectional study. Participants were measured one day before the event, before the first match, and before the second match of the ITF Junior Davis Cup event. Each of the registers lasted 5 min and the time domain, frequency domain and non-linear variables were extracted. The tournament took place in Valencia (Spain). Results showed that there was not a significant decrease in the heart rate variability (HRV) induced by precompetitive stress. However, significant differences (*p*-value < 0.05) were found between baseline and second match pre-competition HRV values in low frequency (LFNu) (*p*-value = 0.008) and high frequency (HFNu) power (*p*-value = 0.008), LF/HF ratio (*p*-value = 0.008) and Sample Entropy (SampEn) (*p*-value = 0.033) variables. Furthermore, cognitive anxiety (measured by CSAI-2r) (*p*-value = 0.017) and anxiety (measured by STAI-S) (*p*-value = 0.042) were higher before the first match than before the second match. Coaches and physical trainers could use HRV to control and manage the precompetitive anxiety of junior tennis players. The experience in international tournaments and the familiarization with the environment in the second match could explain these results.

## 1. Introduction

The heart rate variability (HRV) is a non-invasive index that measures the variation of successive heartbeats over time [1]. This index provides relevant information regarding the sympathetic and parasympathetic nervous system balance. When the HRV is reduced, the sympathetic nervous system has a predominant activity whereas, when the HRV is increased, the parasympathetic nervous system has a predominant innervation. A reduced HRV is related to a reduced capacity to adapt to different environmental challenges [2]. 

HRV can be altered by physical exercises such as strenuous exercise [3], endurance training [4], a padel match [5], or a table tennis match [6]. Thus, HRV is considered an overtraining biomarker [7]. However, changes in HRV are not limited to the impact of physical exercise. Rather, stress, emotions, or even anxiety can alter HRV in athletes [8,9,10,11]. For instance in female soccer players, Ayuso-Moreno, et al. [12] found a reduced HRV before a match. Similar results have been found in swimmers [13,14], mountain bike cyclists [15], or BMX cyclists [16], among other sports, with HRV reductions prior to the competition.

However, the type of sport might influence precompetitive anxiety. Koronas, et al. [17] showed that individual sports such as tennis could have higher precompetitive anxiety than team sports. This could be explained because, in individual sports, all the responsibility for success or failure falls on one athlete [18], which increases feelings of worry, uncertainty, and concern. In this context, a previous study has shown that an excess of precompetitive anxiety could reduce sports performance [19]. The traditional assumption that anxiety is always negative and detrimental to sports performance has been challenged, anxiety symptoms are interpreted as facilitative by elite sports performers, while less skilled counterparts consider anxiety debilitative to performance, indeed, experiencing competitive anxiety can result in positive performance consequences if the individual remains in control [20,21]. Therefore, it may seem crucial that precompetitive anxiety could be managed and controlled in sports contexts to achieve optimal performance.

Research on tennis and precompetitive anxiety has shown consistent results. One study conducted with male and female players showed a precompetitive response with an anticipatory rise of cortisol before a competition [22]. Similar results have been found in young elite female tennis players [23]. Although hormonal assessments were conducted in these studies [22,23], HRV was not registered. In tennis, precompetitive anxiety has been detected in two studies focused on young tennis players [24,25]. Furthermore, precompetitive assessments were registered at baseline, pre-competition, or even post-competition, but not during the tournament in successive matches. This approach has been previously applied to BMX cyclists [16], although to the best of our knowledge, no previous research has studied these assessments on elite international junior tennis players.

Therefore, the present study aimed to investigate the evolution of the HRV and precompetitive anxiety during a tennis tournament among elite international junior tennis players. As previous studies suggested [12,22], we hypothesized that HRV and precompetitive anxiety would decrease before the competition as compared to 24 h before the match. Furthermore, it was also hypothesized that, on the second day of competition, HRV would decrease due to the fatigue induced by the previous tennis match. Since this is the first study during a junior tournament, results would increase the knowledge about the evolution of the precompetitive anxiety and HRV response during an official junior tournament.

## 2. Materials and Methods

### 2.1. Participants 

A total of 18 male elite junior tennis players from 6 different national teams, 3 players representing each country (Slovenia, France, Estonia, Greece, and Spain) were initially recruited before the final of the prestigious men’s team competition 2021 Borotra Cup Tennis Europe U16 (ITF Junior Davis Cup). However, the study did not include 7 participants who did not play at least in two matches. This is because the goal of the study was to investigate the effects of two consecutive matches on the HRV and precompetitive anxiety of the players.

Therefore, a final cohort of 11 elite international junior tennis players was included in the study. Participants had a mean age of 15.36 (0.67) years old and a tennis playing experience of 11 (1.90) years. All the included participants trained more than 150 min/week. Furthermore, they had a mean experience in international competitions of 4.55 (2.25) years. All of them can be considered some of their country’s best players in their category, having been selected by their national federation to represent the country in the world’s most prestigious international team tennis tournament. Therefore, following the classification framework for athletes created by McKay, et al. [26], our tennis players can be considered as tier 4: Elite/International level.

All participants, parents, or legal guardians were informed prior to the enrollment and accepted and gave written consent to all the procedures. The University of Extremadura ethics committee approved the protocol and procedures of the study (number: 112/2021).

### 2.2. Procedure

HRV was collected from the participants at baseline and before the competitions. Thus, five-minute HRV short-term records were conducted at baseline and pre-first match (pM1), and pre-second match (pM2). HRV was assessed at baseline, 24 h before the competition’s first match. In addition, HRV was also assessed five minutes before starting the warm-up of the matches (the first and the second matches). All the measurements were conducted in a calm room with controlled temperature of 20 (1.0) °C. Participants were not allowed to speak during the assessments. In addition, participants were not allowed to take any substance, drink, or drug that may have an impact on the nervous system 24 h before the protocols.

The tournament took place at the Valencia Tennis Center, in Valencia (Spain) from 28 to 30 July 2021. HRV assessments were conducted between 12:00 and 16:00 h. The first and the second matches were played respectively on the first and second day of the competition. 

### 2.3. Instruments and Outcomes

In order to assess the HRV, the Polar RS800CX (Finland) was employed [27]. Recommendation offered by Catai, et al. [28] and the European Society of Cardiology, and the North American Society of Pacing and Electrophysiology [29] were taken into account to assess and report HRV parameters. 

The Kubios HRV software (v. 3.3) [30] was employed to analyze the HRV data. The following different preprocessing steps were applied to assist in this process, such as: (1) a middle filter, which helps to identify those heartbeats intervals (RR intervals), which are shorter/longer than 0.25 s compared to previous beats average, and (2) artifacts were replaced using spline interpolation. After these preprocessing steps, Kubios HRV software provided measures in different domains (frequency, time, and non-linear). 

Regarding time domain, the mean heart rate (mean HR), the proportion of consecutive RR intervals that differ by more than 50 ms (Pnn50), RR intervals, and the square root of differences between adjacent RR intervals (RMSSD). Among the frequency domain variables, high frequency (HF, 0.15–0.4 Hz), the low frequency (LF, 0.04–0.15 Hz), the ratio (LF/HF), and the total power. Lastly, the non-linear measures analyzed in the study were the RR variability from heartbeat to short-term Poincaré graph (width) (SD1), RR variability from heartbeat to long-term Poincaré graph (length) (SD2), and the sample entropy (SampEn).

The Spanish version of the competitive state anxiety inventory-2R (CSAI-2R) was used to evaluate the precompetitive anxiety level [31,32], at baseline, pM1, and pM2. The cognitive anxiety, somatic anxiety, and self-confidence were calculated from the questionnaire’s 17 items (using a 4-point Likert scale). Higher values represented higher cognitive anxiety, somatic anxiety, and self-confidence levels. The Spanish version of the state-trait anxiety inventory (STAI-S) [33] was also used in this study.

### 2.4. Statistical Analysis

The SPSS statistical package (Statistical Package for Social Sciences, version 25) was used to analyze the data. According to Shapiro–Wilk tests, non-parametric analyses were performed. The Friedman test evaluated the HRV differences between baseline, pre-first match (pM1), and pre-second match (pM2) values. Furthermore, pairwise comparisons between conditions were conducted using the Mann–Whitney U test for the HRV and precompetitive anxiety data. Spearman’s Rho correlation were conducted to explore the relationship between HRV and precompetitive anxiety before the two matches. The effect sizes [r] were classified as follows: 0.5 is a large effect, 0.3 is a medium effect, and 0.1 is a small effect [34,35]. The effect size was calculated as follows: $$r=\frac{Z}{\sqrt{N}}$$
where Z is obtained from Mann–Whitney U test and *N* is the total sample. 

## 3. Results

Table 1 shows the characteristics of the participants in the sample.

Table 2 shows the evolution of the HRV during the competition (baseline, pM1, and pM2). Significant differences were not found in any HRV variables (see Table 2).

However, when pairwise comparisons between baseline and pre-matches were conducted, differences emerged in the frequency domain, and the non-linear measures. A significant reduction in the HFnu (*p*-value = 0.008) between the baseline and pM2 was found. LFnu (*p*-value = 0.008) and LF/HF (*p*-value = 0.008) ratio significantly increased between the baseline and pM2. In addition, a similar change was found in the SampEn with a significant reduction (*p*-value = 0.033) between baseline and pM2. 

Table 3 shows the values of precompetitive anxiety obtained in the CSAI-2r and STAI-S questionnaires. Significant differences were found in the cognitive dimension of the CSAI-2r questionnaire (*p*-value = 0.017) with higher values prior to the pM1 as well as in the STAI-S score, again with higher values prior to the pM1 (*p*-value = 0.042).

Table 4 and Table 5 show the correlation between HRV and precompetitive anxiety values obtained in the CSAI-2r and STAI-S questionnaires. Table 4 showed the correlation between HRV and precompetitive values before the first match. Significant correlations were found between SDNN and cognitive anxiety (*p*-value = 0.032; Spearman’s rho = −0.709). Moreover, total power significantly correlated with somatic anxiety (*p*-value < 0.001; Spearman’s rho = −0.976) and STAI-S values (*p*-value = 0.013; Spearman’s rho = −0.783). In addition, SampEn significantly correlated with self-confidence (*p*-value = 0.030; Spearman’s rho = 0.683). Table 5 showed the correlation between HRV and precompetitive values before the second match. Significant correlations were not found.

## 4. Discussion

This paper aimed to investigate the evolution of the HRV during two consecutive matches of an elite international junior tennis tournament. We hypothesized that the precompetitive anxiety would decrease the HRV before the matches. Furthermore, we hypothesized that the pM2 HRV match would decrease due to the physical load experienced in the pM1. However, results did not show statistically significant differences between baseline and pM1 values. In contrast, significant differences were found in the HRV frequency domain and the non-linear measure (SampEn) between baseline and pM2. Regarding the results obtained with the precompetitive questionnaires, elite international junior tennis players showed less anxiety in the pM2 of the competition than in the pM1. Significant correlations were found between total power and somatic anxiety and STAI-S, between SDNN and somatic anxiety, and SampEn and self-confidence.

Previous studies have shown that a sports competition decreases the HRV due to precompetitive anxiety. These results have been found in soccer players [12], tennis players [23,25], swimmers [13,14], mountain bike cyclists [15], and BMX cyclists [16]. Although the means of HRV variables were influenced by the competition (some of the variables showed a reduction in the HRV, which could be compatible with a sympathetic response), values did not reach the significance level. Hypothetically, this could be due to the high number of competitions in that elite international junior athletes usually participate. A previous study showed that the higher the levels of competitive experience, the lower the precompetitive anxiety [36,37]. In this regard, a previous study reported that young tennis players compete in a mean of 15 to 25 tournaments per year [38]. Specifically, the eleven elite junior players analyzed in our study had a mean experience in an international competition of 4.55 (2.25) years, and a tennis playing experience of 11 (1.90) years. This probably would decrease the anxiogenic response to competition.

Research has shown that tennis induces significant psychological stress and physical load [39,40] due to the characteristics of the sport, where intermittent high-intensity efforts are mixed with low-intensity activity and active and passive recoveries [40]. This could be the reason why a significant change in the autonomic modulation (HRV) can be observed between baseline and pM2. In this context, a significant reduction in the HFnu and SampEn, as well as a significant increase in the LFnu and LF/HF ratio, was found. Although controversial interpretations regarding frequency domains and non-linear measures can be found in the literature [41], these results might be explained due to a depressed vagal modulation and a prevalence of sympathetic modulation of the athletes. HF reflects the vagal tone, and LF and LF/HF are a mix of sympathetic and vagal activity [42]. As related to SampEn, it is a non-linear index that measures the complexity of the signal [43,44]. The results showed that complexity is reduced in the aC2, which can be interpreted as a more regular behavior of the organism. A previous study found that SampEn was nearly correlated to cognitive anxiety [16] or influenced by anxiety [45]. In our study, we found a significant correlation between SampEn and self-confidence before the first match. Therefore, future studies should explore the role of this index on precompetitive anxiety levels in sports contexts.

Regarding the precompetitive anxiety levels registered in the questionnaires, our results showed that elite international junior tennis players had lower anxiety (assessed by the STAI-S questionnaire) and lower cognitive anxiety (assessed by the CSAI2r) levels in the pM2 than in the pM1 of the competition. Moreover, significant correlations were found between total power and somatic anxiety and STAI-S as well as between SDNN and cognitive anxiety in the pM1. Similar results were found in BMX cyclists, where somatic anxiety and cognitive anxiety levels were significantly reduced from pM1 to pM2 [16]. The authors suggested that these results could be due to repetitive exposure to precompetitive pressure, which reduced the fear of failure in competition and thus lowered the anxiogenic response of the athletes. Moreover, a previous study showed that athletes’ familiarity with a specific competitive setting influenced their precompetitive anxiety [46]. Thus, all these factors together could act as a buffer to reduce the precompetitive anxiety in pM2 [16]. 

These results found in relation to the precompetitive anxiety levels in youth elite international athletes are interesting for federations, academies, coaches, and physical trainers. Regarding federations, the International Olympic Committee recommended these institutions to monitor the training and competition demands faced by these young athletes because they might be exposed to considerably high levels of psychophysiological stress [47]. Therefore, our results would help to clarify the psychophysiological demands of one of the most important competitions in junior tennis, the ITF junior Davis Cup. This knowledge about the demands of the international competition arena is crucial to improve the training process and the competitive pathway in youth sports [48]. In addition, it would be also recommended for coaches and physical trainers to conduct previous training in the facilities where the competition will take place. This could also reduce the pre-competitive anxiety. 

Despite the fact that this study has followed procedures used in previous research, some limitations should be mentioned. Firstly, the relatively small sample size could cause only greater differences to reach the significance level. Secondly, the sample was comprised of elite international junior male tennis players. This would mean that the results cannot be extrapolated to adult, amateur, female, or junior tennis players. Thirdly, it was not possible to register assessments during and after the matches due to tournament regulations and team availability. Thus, future studies should consider and evaluate these variables in samples of different characteristics and, if possible, during matches to better understand the anxiety processes that tennis players have to deal with. Lastly, sleep data were not available for including it as covariates [26]. Future studies should investigate the relationship between precompetitive anxiety, autonomic modulation, and sleep. Considering these limitations, results may be taken with caution.

## 5. Conclusions

In the first match, elite international junior tennis players perceived higher anxiety levels than in the second match. In addition, higher values of LF and lower values of HF and SampEn in the second match compared to the baseline level could be related to a higher sympathetic modulation due to fatigue.

## Figures and Tables

**Table 1 biology-12-00017-t001:** Descriptive data of participants.

Variables	Mean (SD)
Age (years)	15.36 (0.67)
Tennis playing experience (years)	11 (1.90)
Experience in international competitions (years)	4.55 (2.25)
Height (cm)	181.45 (8.23)
Weight (kg)	66.45 (7.34)

**Table 2 biology-12-00017-t002:** Evolution of the pre-match HRV in young elite international tennis players during the ITF Junior Davis Cup.

Variables	Baseline Mean (SD)	pM1 Mean (SD)	pM2 Match (SD)	*p*-Value	Effect Size
mean HR	75.17 (11.15)	83.67 (7.85)	82.40 (13.26)	0.695	0.219
RR interval	820.95 (126.60)	726.77 (72.16)	750.39 (115.84)	0.695	0.219
pNN50	25.26 (23.14)	11.19 (11.52)	17.27 (14.17)	0.761	0.164
SDNN	53.31 (26.03)	40.26 (10.07)	48.24 (17.14)	0.695	0.219
RMSSD	50.80 (34.46)	31.41 (12.29)	37.60 (20.30)	0.695	0.219
HFnu	32.70 (16.96)	25.62 (12.59)	18.60 (9.04)	0.178	1.041
LFnu	67.21 (17.00)	74.23 (12.70)	81.35 (9.05)	0.178	1.041
LF/HF	3.10 (2.47)	4.06 (2.76)	5.87 (3.80)	0.178	1.041
Total power	2963 (2978)	1413 (749)	2330 (1538)	0.695	0.219
SD1	35.97 (24.41)	22.24 (8.71)	26.62 (14.38)	0.695	0.219
SD2	65.61 (29.34)	52.13 (12.73)	62.54 (20.65)	0.913	0.054
SampEn	1.73 (0.27)	1.55 (0.19)	1.50 (0.40)	0.060	1.700

pM1: pre-first match; pM2: pre-second match. HR: heart rate; RR: time between intervals R-R; RMSSD: the square root of the mean of the squares of the successive differences of the interval RR; pNN50: percentage of intervals >50 ms different from the previous interval; total power: the sum of all the spectra; LF: low frequency (ms2); HF: high frequency (ms2); SampEn: sample entropy; SD1: dispersion, standard deviation, of points perpendicular to the axis of line-of-identity in the Poincaré plot; SD2: dispersion, standard deviation, of points along the axis of line-of-identity in the Poincaré plot.

**Table 3 biology-12-00017-t003:** Precompetitive anxiety values before the first and the second match of the ITF Junior Davis Cup.

Variables	pM1 Mean (SD)	pM2 (SD)	*p*-Value	Effect Size
Cognitive	1.85 (0.58)	1.38 (0.53)	0.017	0.721
Somatic	1.54 (0.41)	1.56 (0.84)	0.527	0.190
Self-confidence	3.34 (0.58)	3.40 (0.61)	0.750	0.136
STAI-S	36.70 (9.75)	31.12 (9.72)	0.042	0.613

pM1: pre-first match; pM2: pre-second match; STAI-S: state-trait anxiety inventory.

**Table 4 biology-12-00017-t004:** Correlation between HRV and precompetitive values before the first match.

	Mean HR	RR Interval	pNN50	SDNN	RMSSD	HFnu	LFnu	LF/HF	Total Power	SD1	SD2	SampEn
Cognitive	0.582	−0.582	−0.633	−0.709 *	−0.641	−0.169	0.169	0.169	−0.388	−0.641	−0.591	−0.110
Somatic	0.610	−0.610	−0.512	−0.610	−0.342	0.293	−0.293	−0.293	−0.976 *	−0.342	−0.610	0.171
Self-confidence	−0.335	0.335	0.287	0.220	0.122	−0.055	0.055	0.055	0.506	0.122	0.098	0.683 *
STAI-S	0.617	−0.617	−0.550	−0.500	−0.400	−0.033	0.033	0.033	−0.783 *	−0.400	−0.317	−0.517

* *p*-value < 0.05.

**Table 5 biology-12-00017-t005:** Correlation between HRV and precompetitive values before the second match.

	Mean HR	RR Interval	pNN50	SDNN	RMSSD	HFnu	LFnu	LF/HF	Total Power	SD1	SD2	SampEn
Cognitive	−0.696	0.696	0.696	0.609	0.609	0.609	−0.609	−0.609	0.348	0.609	0.348	0.464
Somatic	0.116	−0.116	−0.116	0.087	0.087	0.087	−0.087	−0.087	−0.145	0.087	−0.145	0.348
Self-confidence	−0.100	0.100	0.100	0.100	0.100	0.100	−0.100	−0.100	0.500	0.100	0.500	−0.100
STAI-S	−0.200	0.200	0.200	0.200	0.200	0.200	−0.200	−0.200	−0.400	0.200	−0.400	0.400

## Data Availability

Data will be available upon reasonable request to the corresponding author.

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
