# Peer review of "Heart Rate Variability in Elite International ITF Junior Davis Cup Tennis Players"

_biology, 2022, doi:10.3390/biology12010017_

Round 1

Reviewer 1 Report

Dear Authors,

It has been a pleasure to read your manuscript aimed to investigate the evolution of the HRV during a tennis tournament in elite international junior tennis players. Results of the study could be of interest for stakeholders.

Nonetheless, in my opinion some clarifications are needed in the results and discussion section.

My major concerns are that some critical aspects of the facilitatory or inhibitory effects of competitive anxiety have not been addressed. Also, further details are needed in the description of the procedures, and the conclusions seems to report contradicting findings.

I also included some suggestions that you could consider to strengthen your results.

Minor

Line 2: in the tile I would prefer to avoid acronyms and express the complete definition of Heart rate variability.

Line 47: the expression “acute maneuvers” should be better explained.

Line 74: what is meant by “the physical impact” should be explained.

Line 84: competitive level of athletes could be better defined as illustrated in the following reference: McKay AKA, Stellingwerff T, Smith ES, Martin DT, Mujika I, Goosey-Tolfrey VL, Sheppard J, Burke LM. Defining Training and Performance Caliber: A Participant Classification Framework. Int J Sports Physiol Perform. 2022 Feb 1;17(2):317-331. doi: 10.1123/ijspp.2021-0451.

Lines 133-136: these results are already presented in the table and partially in the “2.1. Participants” section, please avoid redundancy.

Major

Lines 57-58: here you should address the very important issue of facilitatory or inhibitory effect of anxiety on sport performance, hence: The traditional assumption that anxiety is always negative and detrimental to sport performance has been challenged, anxiety symptoms are interpreted as facilitative by elite sport performers, while less skilled counterparts consider anxiety debilitative to performance, indeed, experiencing competitive anxiety can result in positive performance consequences if the individual remains in control. On this topic see: Demarie, S., Minganti, C., Piacentini, M. F., Parisi, A., Cerulli, C., & Magini, V. (). Reducing anxiety in novel horse riders by a mechanical horse simulator. Med Sport. 2013; 66:179-188. AND Jones G, Hanton S. Pre-competitive feeling states and directional anxiety interpretations. J Sports Sci. 2001; 19(6):385-95. doi: 10.1080/026404101300149348

Lines 72-73: HRV before the competition was compared 24 h before the match, it does not seem correct to mention the comparison with the day of the match.

Line 92: the time of day of HRV collection should be specified.

Line 96: warm-up protocol should be described, and a timeline of all baseline measurements should be specified accordingly. i.e. were they all taken before or after war-up?

Lines 107 and 111: RR and RR50 should be defined.

Lines 171-172: to substantiate your findings you should describe to how many tournaments your subjects participated per year.

Line 215: please add that your results cannot be extrapolated to females.

Lines 222-224: your conclusion sees contradicting. You report less anxiety levels in the second match, but differences between baseline and precompetitive HRV only in the second match. You should better address these findings.

Further comments

Lines 197-188: “A previous study found that SampEn was nearly correlated to cognitive anxiety [15] or influenced by anxiety [36]”. It looks like you have those data, so, maybe, adding the analysis yourself could strengthens your results. Also, the analysis of the correlation among HRV and questionnaires results could add further value to your work.

Line 199: some suggestion on the practical application of the athletes’ familiarity with a specific competitive setting could be of interest for the reader.

Author Response

Dear Authors,

It has been a pleasure to read your manuscript aimed to investigate the evolution of the HRV during a tennis tournament in elite international junior tennis players. Results of the study could be of interest for stakeholders.

Nonetheless, in my opinion some clarifications are needed in the results and discussion section.

My major concerns are that some critical aspects of the facilitatory or inhibitory effects of competitive anxiety have not been addressed. Also, further details are needed in the description of the procedures, and the conclusions seems to report contradicting findings.

I also included some suggestions that you could consider to strengthen your results.

 Author´s response: Thank you for all your constructive and valuable comments. After considering all of them, we truly believe that the quality of the manuscript has been significantly improved.

Minor

Line 2: in the tile I would prefer to avoid acronyms and express the complete definition of Heart rate variability.

Author´s response: Corrected.

Line 47: the expression “acute maneuvers” should be better explained.

Author´s response: Thank you for your comment. It was a typo and it has been corrected.

Line 74: what is meant by “the physical impact” should be explained.

Author´s response: Thank you for your comment. You were right that physical impact might be ambiguous. Thus, we have rewritten the sentence and included that HRV “would decrease due to the fatigue induced by the previous tennis match”.

Line 84: competitive level of athletes could be better defined as illustrated in the following reference: McKay AKA, Stellingwerff T, Smith ES, Martin DT, Mujika I, Goosey-Tolfrey VL, Sheppard J, Burke LM. Defining Training and Performance Caliber: A Participant Classification Framework. Int J Sports Physiol Perform. 2022 Feb 1;17(2):317-331. doi: 10.1123/ijspp.2021-0451.

Author´s response: Thank you for this interesting and valuable article. We have used it to classify our participants as well as to include some missing details in the description of the sample.

Lines 133-136: these results are already presented in the table and partially in the “2.1. Participants” section, please avoid redundancy.

Author´s response: Thank you for your comment. Following your recommendation, these lines have been removed.

Major

Lines 57-58: here you should address the very important issue of facilitatory or inhibitory effect of anxiety on sport performance, hence: The traditional assumption that anxiety is always negative and detrimental to sport performance has been challenged, anxiety symptoms are interpreted as facilitative by elite sport performers, while less skilled counterparts consider anxiety debilitative to performance, indeed, experiencing competitive anxiety can result in positive performance consequences if the individual remains in control. On this topic see: Demarie, S., Minganti, C., Piacentini, M. F., Parisi, A., Cerulli, C., & Magini, V. (). Reducing anxiety in novel horse riders by a mechanical horse simulator. Med Sport. 2013; 66:179-188. AND Jones G, Hanton S. [1]. J Sports Sci. 2001; 19(6):385-95. doi: 10.1080/026404101300149348

Author´s response: Thank you for providing us these interesting references. We have followed your valuable suggestion and included it into the introduction.

Lines 72-73: HRV before the competition was compared 24 h before the match, it does not seem correct to mention the comparison with the day of the match.

Author´s response: You were totally right. We have corrected this typo as follows: “…would decrease before the competition as compared to 24h before the match.”

Line 92: the time of day of HRV collection should be specified.

Author´s response: Done.

Line 96: warm-up protocol should be described, and a timeline of all baseline measurements should be specified accordingly. i.e. were they all taken before or after war-up?

Author´s response: Thank you for your comment. Since HRV is significantly affected by physical exercise, HRV assessments were recorded 5 min before starting the warm-up. We have specified this in the manuscript to avoid misunderstandings. In relation with that, the warm-up was not standardized since it is slightly different depending on the national team.

Lines 107 and 111: RR and RR50 should be defined.

Author´s response: Done.

Lines 171-172: to substantiate your findings you should describe to how many tournaments your subjects participated per year.

We agree with the reviewer that this could be interesting to know, but unfortunately, we did not include it in the data requested from the players. Moreover, players of this category and level play tournaments regulated by different organisations (ITF, Tennis Europe, National Federations, etc.) which are not always published on the internet, so it is not possible to check the number of tournaments played a posteriori. However, we believe that this data is not of excessive relevance since, as described in the section on participants, all of them are players who can be categorised as elite or international level and, therefore, the sample is homogeneous.

Line 215: please add that your results cannot be extrapolated to females.

Author´s response: Done.

Lines 222-224: your conclusion sees contradicting. You report less anxiety levels in the second match, but differences between baseline and precompetitive HRV only in the second match. You should better address these findings.

Author´s response: Thank you for your comment. Following your suggestion, we have modified the conclusions.

Further comments

Lines 197-188: “A previous study found that SampEn was nearly correlated to cognitive anxiety [15] or influenced by anxiety [36]”. It looks like you have those data, so, maybe, adding the analysis yourself could strengthens your results. Also, the analysis of the correlation among HRV and questionnaires results could add further value to your work.

Author´s response: Thank you for your suggestion. Correlations have been included.

Line 199: some suggestion on the practical application of the athletes’ familiarity with a specific competitive setting could be of interest for the reader.

Author´s response: Thank you for your comment. We have added this information.

Reviewer 2 Report

I am grateful for the opportunity to review this manuscript titled “HRV in elite international ITF junior Davis Cup tennis players”. The purpose of this study was to investigate the evolution of the HRV during an international team tennis tournament in elite junior tennis players.

This study is of interest to BIOLOGY readers and seems to provide some new findings, applicable to the fields of training. However, the points mentioned in the “Specific comments” section below should be considered and the manuscript amended accordingly before being considered for publication. Also, a native English speaker should read and review the manuscript before its submission.

Specific comments

Title

1.     I recommend changing the title and avoiding the use of abbreviations.

Abstract

2.     The authors could briefly describe the protocols.

3.     It would be appropriate for authors to introduce statistical values in the abstract (i.e., p-value, effect size, ...).

Introduction

4.     The introduction is well written, but the authors need to highlight in the introduction the contribution of their work to the area.

Materials and Methods

2.1. Participants

5.     The authors should mention whether the participants were only men, or whether the sample was also composed of women.

2.2. Procedure

6.     Could the authors describe how HRV was measured in more detail?

7.     Was sleep considered as a covariate? This can also alter HRV.

2.4. Statistical analysis

8.     How was the effect size obtained?

Results

9.     Line 140: Please describe HR, RR, pNN50, SDNN, SDNN, RMSSD, ...

10.  Line 151: Please describe STAI-S.

Conclusions

11.  The conclusion is repetitive and ambiguous. I recommend the authors to rewrite this section.

Author Response

I am grateful for the opportunity to review this manuscript titled “HRV in elite international ITF junior Davis Cup tennis players”. The purpose of this study was to investigate the evolution of the HRV during an international team tennis tournament in elite junior tennis players.

This study is of interest to BIOLOGY readers and seems to provide some new findings, applicable to the fields of training. However, the points mentioned in the “Specific comments” section below should be considered and the manuscript amended accordingly before being considered for publication. Also, a native English speaker should read and review the manuscript before its submission.

Author´s response: Thank you for all your constructive and valuable comments. After considering all of them, we truly believe that the quality of the manuscript has been significantly improved.

Specific comments

Title

  1. I recommend changing the title and avoiding the use of abbreviations.

 Author´s response: Done.

Abstract

  1. The authors could briefly describe the protocols.

Author´s response: Done.

  1. It would be appropriate for authors to introduce statistical values in the abstract (i.e., p-value, effect size, ...).

 Author´s response: Done.

Introduction

  1. The introduction is well written, but the authors need to highlight in the introduction the contribution of their work to the area.

Author´s response: Thank you for your kind recommendation. We have included it after the hypotheses.

Materials and Methods

2.1. Participants

  1. The authors should mention whether the participants were only men, or whether the sample was also composed of women.

Author´s response: Included.

2.2. Procedure

  1. Could the authors describe how HRV was measured in more detail?

Author´s response: Included.

  1. Was sleep considered as a covariate? This can also alter HRV.

Author´s response: Thank you for your comment. Unfortunately, we have no access to participants´ sleep data. We have included this limitation in the limitation section.

2.4. Statistical analysis

  1. How was the effect size obtained?

 Author´s response: Done.

Results

  1. Line 140: Please describe HR, RR, pNN50, SDNN, SDNN, RMSSD, ...

Author´s response: Included.

  1. Line 151: Please describe STAI-S.

Author´s response: Included.

Conclusions

  1. The conclusion is repetitive and ambiguous. I recommend the authors to rewrite this section.

Author´s response: Done.

Reviewer 3 Report

First of allmy most sincere congratulations for the interesting and necessary researchwhich contributes to a greater knowledge of young tennis players from different countries. HoweverI consider thatabove allthe objective, hypotheses and conclusions are intuitive , in addition to observing aspects that need to be improved. In this sense, I hope and wish always the best, being available for any question

Author Response

First of all, my most sincere congratulations for the interesting and necessary research, which contributes to a greater knowledge of young tennis players from different countries. However, I consider that, above all, the objective, hypotheses and conclusions are intuitive , in addition to observing aspects that need to be improved. In this sense, I hope and wish always the best, being available for any question.

Author´s response: Thank you for your positive and constructive feedback. We have modified the objectives, hypotheses and conclusion sections in order to be more accurate. Please, do not hesitate if there are any other change that could improve the quality of the manuscript.

Reviewer 4 Report

First of all, I would like to thank you for the opportunity to review this study. I certainly consider the study to be very well-founded from a theoretical and methodological perspective. However, before accepting it, I believe that a few modifications are necessary. 

Regarding the introduction, this is very well founded, however, I think that more current quotations need to be added. 

Regarding the scope of the participants, I think that more data related to the characteristics of the participants needs to be added - is this possible? Also, has this research been supervised by an ethics committee? Please add this in the procedure section

Add the authors' contributions in the corresponding section. 

Finally, check the bibliography as the references are poorly referenced. 

Author Response

First of all, I would like to thank you for the opportunity to review this study. I certainly consider the study to be very well-founded from a theoretical and methodological perspective. However, before accepting it, I believe that a few modifications are necessary.

Author´s response: Thank you for all your constructive and valuable comments. After considering all of them, we truly believe that the quality of the manuscript has been significantly improved.

Regarding the introduction, this is very well founded, however, I think that more current quotations need to be added.

Author´s response: Thank you for your constructive comment. Some recent articles have been included in this section.

Regarding the scope of the participants, I think that more data related to the characteristics of the participants needs to be added - is this possible? Also, has this research been supervised by an ethics committee? Please add this in the procedure section

Author´s response: Thank you for your comment. We have included further information regarding participants. In this regard, we have classified our sample according to the framework proposed by McKay. In addition, we have highlighted the approval number of the ethics committee.

Add the authors' contributions in the corresponding section.

Author´s response: Done.

Finally, check the bibliography as the references are poorly referenced. 

Author´s response: Thank you for your recommendation. References are included with the EndNote software adapted to the Biology style. Nevertheless, we have corrected some issues.

Round 2

Reviewer 1 Report

Dear Authors,

I congratulate you on responding to all my comments, in the present form the work is well structured and presented.

Reviewer 2 Report

All modifications have been made by the authors. The manuscript is well structured and presented.